# Trend Analysis of Temperature and Precipitation Extremes during Winter Wheat Growth Period in the Major Winter Wheat Planting Area of China

**Hanjiang Nie [1,2]**, **Tianling Qin [2,*]**, **Hanbo Yang [1]**, **Juan Chen [2]**, **Shan He [2]**, **Zhenyu Lv [1,2]** and **Zhenqian Shen [2,3]**

[1] Department of Hydraulic Engineering, Tsinghua University, Beijing 100084, China; nhj199008@163.com (H.N.); yanghanbo@tsinghua.edu.cn (H.Y.); LVZYIWHR@163.com (Z.L.)
[2] State Key Laboratory of Simulation and Regulation of Water Cycle in River Basin, China Institute of Water Resources and Hydropower Research, Beijing 100038, China; chenjuan@iwhr.com (J.C.); hsiwhr61@163.com (S.H.); shenzqbin@163.com (Z.S.)
[3] School of Water Conservancy, North China University of Water Resources and Electric Power, Zhengzhou 45045, China
[*] Correspondence: qintl@iwhr.com; Tel.: +86-10-6878-1316

**Abstract:** In this study, the major winter wheat planting area of China is selected as the study area, with the time scale of the growth period of winter wheat (a total of 56 growth periods during October 1961 to May 2016). The significance, stability, magnitude of the trend and the average trend of the study area with eight temperature indices and seven precipitation indices of 453 meteorological stations are tested by Mann–Kendall method and Sen's nonparametric method. The following observation can be made: (1) the cold extreme indices show strong and stable downward trend in most of the stations in the study area, while the hot extreme indices show a strong and stable upward trend, especially in the northern winter wheat planting area and the north of the southern winter wheat planting area. (2) The trends of extreme precipitation indices in most of the sites in the study area are insignificant and unstable. Only in R20mm, a significant and stable decreasing trend is shown in some stations, which is mainly located in the northern winter wheat planting area and part of the central and western regions in the study area. The results in some ways could enrich the references for understanding the climate change in the growth period of winter wheat in the region and help to formulate a better agronomic management practice of winter wheat.

**Keywords:** climate extremes; Mann–Kendall method; Sen's nonparametric method; growth period; winter wheat

## 1. Introduction

In the past century, the global climate has shown a significant warming trend, and the mean surface temperature has increased by 0.85 °C during 1880 and 2012 [1]. The intensity, frequency, and duration of extreme temperatures and precipitation events in some areas have changed. The natural ecosystems and human activities, such as agricultural production, forest growth, urban planning, water resources management, and human health, have been affected [1–5]. Therefore, to study the change rule of extreme temperature and precipitation was gradually becoming one of the hottest topics in climate change research.

In China, the changes in extreme climate are uneven both in space and time because of the large landmass and various climatic factors [6], and the extreme climate changes are crucial to the local population, economy and environment [7]. The change rules of extreme climate changes in different basins in China have been studied [8–12]. On the whole, the temperature of the five major

basins (Yangtze River Basin, Yellow River Basin, Huai River Basin, Haihe River Basin, and Zhujiang River Basin) in China has shown an upward trend, but precipitation trends in different basins vary. The rainfall days in most parts of the Yangtze River Basin decreased significantly, while the precipitation intensity increased [13]. A decreasing trend in precipitation was shown in the Yellow River Basin [11], Haihe River Basin [12], and Zhujiang River Basin [9]. The trends of annual maximum rainfall are not obvious in Huaihe River Basin [8]. The changes in extreme climates at different time scales have also been studied. For example, the summer rainfall showed a significant upward trend in southern China and a significant downward trend in northern China, which have been found by Ye [14]. However, the trend of the annual precipitation was not significant.

Climate is the major uncontrollable factor in crop yields. It has also been pointed out that the rate of increase in food crop yields has slowed because of climate change [15]. Several studies have shown that crop yields have been decreasing with the temperature increasing since the 1980s due to the accelerating growth of the phenophase, shortening the duration of crop growth and grain filling period, aggravating heat-related water stress, and exacerbating pest and disease losses [16,17]. Further studies also showed that the continued increase in mean annual temperature and extreme high temperature can reduce the potential amount of grains and the number of seeds [18,19]. In addition, the growth of crops can be influenced by soil moisture by affecting the root respiration. When the soil moisture is excessive because of heavy precipitation, the root respiration would weaken and even stop [20]. Great damage to crop production was caused by heavy rainfall events around the world in the past several years. If the frequency of extreme precipitation increases in the future, the result may be even worse [21,22]. Therefore, studying the changing patterns of extreme temperatures and precipitation during the growth period of crops is a target to ensure the future crop yield.

Wheat is the largest, most productive and widely distributed grain crop in the world, and wheat production accounts for 1/3 of the world's total food production. China is the largest producer of wheat in the world, while winter wheat accounts for 98% of the total wheat yield in China [23]. In this paper, the Chinese winter wheat planting area (Figure 1) is chosen to identify the spatial and temporal distribution of temperature and precipitation extremes during the winter wheat growth period. This would help to guide the field management of winter wheat in the region and provide certain support for the protection of winter wheat production in response to future climate change.

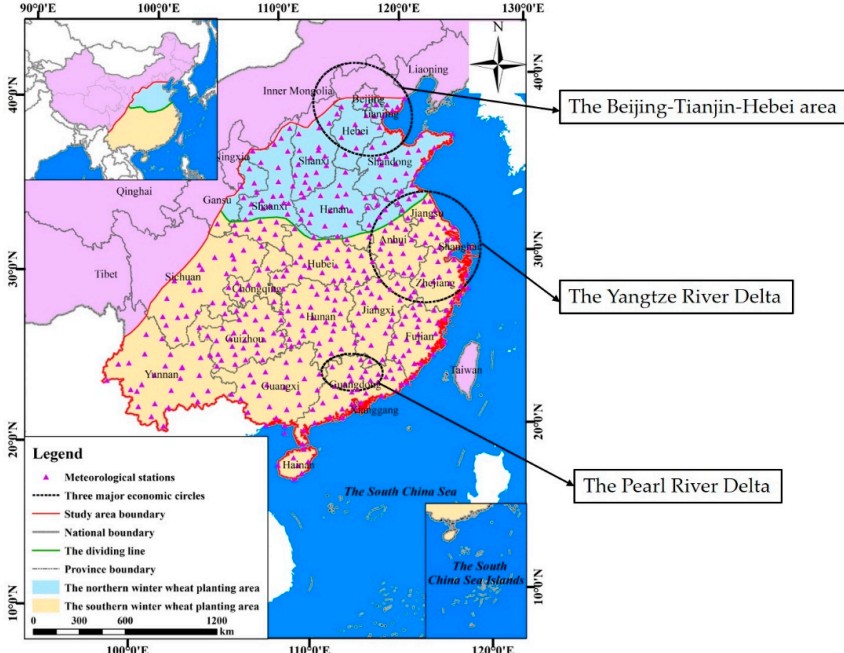

**Figure 1.** Distribution map of winter wheat planting areas and meteorological stations in China.

## 2. Materials and Methods

### 2.1. Study Area and Data

The winter wheat planting area of China can be divided into the northern winter wheat planting area and the southern winter wheat planting area (Figure 1). The northern and southern winter wheat planting areas are separated by the Qinling Mountains and Huaihe River [24,25]. The study area involves 23 provinces with a total area of 3.225 million km$^2$, including Beijing, Tianjin, Hebei, Shandong, Henan and Anhui. China's three major economic circles (i.e., the Yangtze River Delta, the Pearl River Delta, and the Beijing–Tianjin–Hebei area) are also included in the area. In 2017, the regional winter wheat production was 127.35 million tons, accounting for 98% of China's total wheat yield (including winter wheat and spring wheat) and 21% of China's total grain yield [23]. Therefore, ensuring winter wheat yield in this region is crucial to China's food security.

### 2.2. Temperature and Precipitation Data

According to the continuous and integrated meteorological station data series, the data of daily precipitation, daily mean temperature, daily maximum temperature and daily minimum temperature from 1961 to 2016 at 487 meteorological stations in the study area used were provided by the National Meteorological Information Center of China Meteorological Administration (available at http://data.cma.cn/). Due to the unavoidable occurrence of unreasonable data and missing values in the data, it is necessary to carry out quality control and homogeneity test on the data [26].

### 2.3. Data Quality and Homogeneity

In the analysis of long sequence data, it is necessary to test the quality of data. In this paper, the RClimdex software [27] was used to test it. The RClimdex software has been used by many researchers, and it can be applied to the quality analysis of meteorological data [6,28,29]. The following two aspects are modified by the software: (1) the outliers in the daily precipitation, daily average temperature, daily minimum temperature, and daily maximum temperature data were identified. Moreover, the outside ranges were defined as lying beyond four standard deviations (std) of the mean value (mean ±4 × std) [28,30]. (2) The unreasonable data mainly included data that the daily minimum temperature is greater than the daily maximum temperature and the daily precipitation is negative. The processing methods for the above two data problems are: (1) for outlying data, the data could be set to the multi-year average of the same day. (2) The unreasonable data were set as the missing value, and the missing values should not be involved in the calculation of the extreme climatic index. Finally, the data were tested again using the RClimDex software to ensure that there were no abnormal data.

Data homogeneity is another element which should be considered in climate data, especially when these data are used to assess climate change [6]. When the weather stations record the longtime series climate data, it is inevitable that the non-climatic factors of the data will be mutated due to the changing of the instruments, observers, sites location or the surrounding environments [31]. In this paper, the RHtests V4 [32] software was used to test the homogeneity of precipitation and temperature data. However, due to the noise in the daily data, statistical analysis is very complicated. For this reason, monthly data were obtained by adding the daily data and then the homogeneity of monthly data was checked by RHtests V4. If the data for a particular month exceeded the 95% confidence interval, the data of the corresponding months were classified as change points and all the daily data in this month were set missing. Finally, the processed data were tested again by using RHtests V4 software to ensure that the data did not have a change point.

Through data quality checks, we identified less than 0.2% observations per station as outliers, and these values were set to the multi-year average of the same day. On average, we corrected approximately 0.8%, 1.2%, and 2.3% of observations per station for maximum temperature, minimum temperatures, and precipitation, respectively. Through the homogeneity check of the data, 121 change points were detected among 487 stations in the monthly data series. All of the change points occurred

between 1978 and 1980. And all the daily data in the change point were set missing. The stations with more than 5% of data missing were excluded from the analysis [26,33]. After data quality and homogeneity checking, 34 stations with missing more than 5% of data were excluded and 453 out of 487 stations were thus included in this study.

In China, the growing season of winter wheat is normally from October to May [24]. Therefore, the data from January to September 1961, from June to September of 1961–2015, and after June 2016 were deleted from this study.

## 2.4. Indices of Extremes

Climate extremes indices recommended by the Expert Team on Climate Change Detection and Indices (ETCCDI) [34] were used to evaluate climate change in many parts of the world [6,35–37]. According to the climatic characteristics of the study area and the biological characteristics of winter wheat, 15 extreme climate indices (Table 1) were selected, including eight extreme temperature indices and seven extreme precipitation indices. The definitions of the 15 extreme indices here are different from the definitions in ETCCDI. The original indices were calculated for the data of one year, and we calculated the indices during the growth period of winter wheat here. For example, the FD0 index of 2015 was calculated. The original index was defined as the number of days in which the minimum temperature is less than 0 °C in 2015, and we defined it as the number of days with the minimum temperature of less than 0 °C in the winter wheat growth period (i.e., October 2015 to May 2016) here. The annual scale index could be directly calculated by RClimdex software [6], but RClimdex software could not calculate other time scale indices. Therefore, this article was programmed with MATLAB R2014a software, which realized the calculation of the extreme index on the growth period scale. The period of the growth period of October 1971 to May 2001 was chosen as the base period for calculation at a station.

**Table 1.** List of the extreme temperature and precipitation indices used in this study.

| Indices | Name | Definitions | Units |
|---|---|---|---|
| Temperature | | | |
| FD0 | Frost days | WWGP (winter wheat growth period) count when TN (daily minimum) < 0 °C | days |
| GSL | Growing season length | WWGP (1st Jan to 31st Dec in NH, 1st July to 30th June in SH) count between the first span of at least six days with TG (daily average) > 5 °C and first span after July 1 (January 1 in SH) of six days with TG < 5 °C | days |
| TXx | Max Tmax | Monthly maximum value of daily maximum temp | °C |
| TNn | Min Tmin | Monthly minimum value of daily minimum temp | °C |
| TN10p | Cool nights | Percentage of days when TN < 10th percentile of October 1971 to May 2001 [1] | % |
| TX10p | Cool days | Percentage of days when TX (daily maximum) < 10th percentile of October 1971 to May 2001 | % |
| TN90p | Warm nights | Percentage of days when TN > 90th percentile of October 1971 to May 2001 | % |
| TX90p | Warm days | Percentage of days when TX > 90th percentile of October 1971 to May 2001 | % |
| Precipitation | | | |
| Rx1day | Max one-day precipitation amount | Monthly maximum one-day precipitation | mm |
| Rx5day | Max five-day precipitation amount | Monthly maximum consecutive five-day precipitation | mm |
| SDII | Simple daily intensity index | WWGP total precipitation divided by the number of wet days (defined as RR (daily precipitation) >= 1.0mm) in the year | mm/day |
| R20mm | Number of days above 20 mm | WWGP count of days when RR > = 20mm | days |
| CDD | Consecutive dry days | Maximum number of consecutive days with RR < 1mm | days |
| R95pTOT | Very wet days | WWGP total PRCP when RR > 95th percentile | mm |
| PRCPTOT | WWGP total wet day precipitation | WWGP total PRCP in wet days | mm |

[1] The period of October 1971 to May 2001 is the base period.

### 2.5. Methods

#### 2.5.1. Mann–Kendall Method

The Mann–Kendall (M–K) trend test [38,39] recommended by the World Meteorological Organization (WMO) is an effective tool to evaluate the trend of hydrometeorological data, such as temperature, precipitation, and runoff [40–43]. In this paper, the M–K test was applied to analyze the trend of temperature and precipitation extremes indices. In order to avoid the results of the trend analysis being affected by using different time periods, we used a 30-growth-period moving window which slided every one growth period during the study period of October 1961 to May 2016 (There are a total of 26 30-growth-period moving windows).

Trends were identified as statistically significant when they were significant at $p \leq 0.1$, but trends at $p \leq 0.2$ were also taken into consideration and designated as weak trends [44,45]. Therefore, the classification criteria of the trend of extreme climate indices can be divided by referring to relevant studies [6,28]: Very strong trends: $p \leq 0.05$. Strong trends: $0.05 < p \leq 0.1$. Weak trends: $0.1 < p \leq 0.2$. Insignificant trends: $p > 0.2$. ($p$ is the tested significance level)

#### 2.5.2. Trend Percentage and Stability

The percentage of statistically significant trends in each extreme index can be calculated by the approach of Lupikasza [44]:

$$PCT_i = \frac{N_{pi}}{N_T \times M} \times 100\% \tag{1}$$

where $PCT_i$ is the percentage of one type of trend strength for a certain index i, $N_T$ is the total number of meteorological stations, $M$ is the total number of the 30-growth-period moving periods (26 movable 30-growth-period during October 1961 to May 2016) and $N_{pi}$ is the number of one type of trend strength that was tested from the number of possible cases ($N_T \times M$).

In general, the trend of hydrological or meteorological sequence data is usually detected on the basis of a specific time period, and the test results are strongly influenced by the selected time periods [6,44]. For the same elements, it is likely that different trends result from different time periods, which reflects the instability of the trend [6]. In order to eliminate the instability of the trend test, the trend of the 30-growth-period (26 movable 30-growth-period during October 1961 to May 2016) of extreme climate indices was calculated by M–K trend test. The stability of such trends is expressed as a percentage of the 30-growth-period during which these trends were statistically significant ($p \leq 0.1$). That is to say, we needed to count the number of very strong upward (or downward, $p \leq 0.05$) and strong upward (or downward, $0.05 < p \leq 0.1$) trends in the 30-growth-period sequence, and then calculate the percentage of them in the total number M. The trend stability of each extreme index can be determined by the following equation [44]:

$$S_{ij} = \frac{M_{1ij} + M_{2ij}}{M} \times 100\% \tag{2}$$

where $S_{ij}$ is trend stability at meteorological station j for index $i$, and $M_{1ij}$ and $M_{2ij}$ are the numbers of the 30-growth-period moving periods with respectively very strong upward (or downward, $p \leq 0.05$) and strong upward (or downward, $0.05 < p \leq 0.1$) trends at station $j$ for index $i$.

The stability of the trend can be divided according to the following criteria [28]: Unstable trends: $0\% \leq S < 25\%$, stable trends: $25\% \leq S < 60\%$, strongly stable trends: $S \geq 60\%$.

According to the above criteria, for a station, if it was ultimately judged as 'stable downward trend', this means that downward trend frequently occurs in the total number of the 30-growth-period moving periods for the station.

2.5.3. Sen's Slope Estimator

The Sen's slope estimator [46] was used to estimate the trend magnitudes in the extreme index series. The slope estimates of all pairs of data were computed in each 30-growth-period moving period:

$$Sen_k = \frac{x_z + x_y}{z - y} \tag{3}$$

where $x_z$ and $x_y$ are data values at times $z$ and $y$ ($z > y$, $z$ and $y \in n$), $n$ is the number of time series, and $k$ is the number of slopes. The median of these $N$ ($N = n(n-1)/2$) values of $Sen_k$ is Sen's estimator of slope. The $N$ values of $k$ are ranked from the smallest to the largest and the Sen's estimator is:

$$Sen = Sen_{[(N+1)/2]}, \text{if } N \text{ is odd} \tag{4}$$

$$Sen = \left(Sen_{[N/2]} + Sen_{[(N+2)/2]}\right)/2, \text{if } N \text{ is even} \tag{5}$$

According to relevant research, the slope computed by Sen's slope estimator is a robust estimate of the magnitude of a trend, which has been widely used in identifying the slope of the trend line in hydrological and meteorological time series [28,47–49].

## 3. Results

### *3.1. Trend Strengths in Extreme Temperature and Precipitation Indices*

3.1.1. Percentage of the Trend Strengths

The percentages of different trend strengths (PCT) for extreme temperature and precipitation indices of the growth period during October 1961 to May 2016 are calculated and graphed in Figures 2 and 3, respectively.

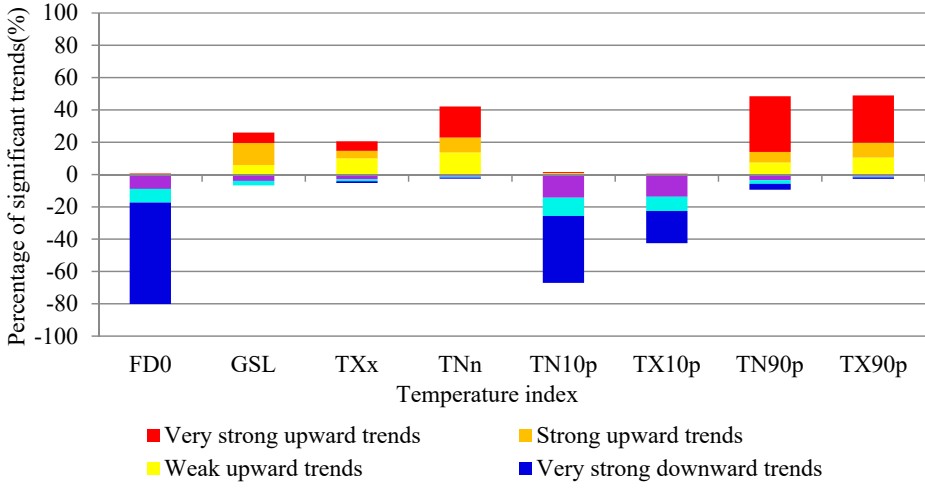

**Figure 2.** Percentages of trend (PCT) with different significant levels for each extreme temperature index in the study area during the growth period of October 1961 to May 2016.

As shown in Figure 2, the downward trends in cold extreme indices (FD0, TN10p, and TX10p) are dominated during the growth period of October 1961 to May 2016 in the study area. Furthermore, decreasing trends occur frequently, with approximately 80% in FD0 (63% of very strong downward trends), 67% in TN10p (41% of very strong downward trends) and 43% in TX10p (20% of very strong downward trends). Hot extreme indices (TXx, Tn90p, and Tx90p) and the other cold extreme index (TNn) indicate general positive trends in the period, with approximately 21% in TXx, 48% in TN90p,

49% in TX90p, and 42% in TNn. For GSL, 26% show an upward trend, with only 7% show very strong upward trends.

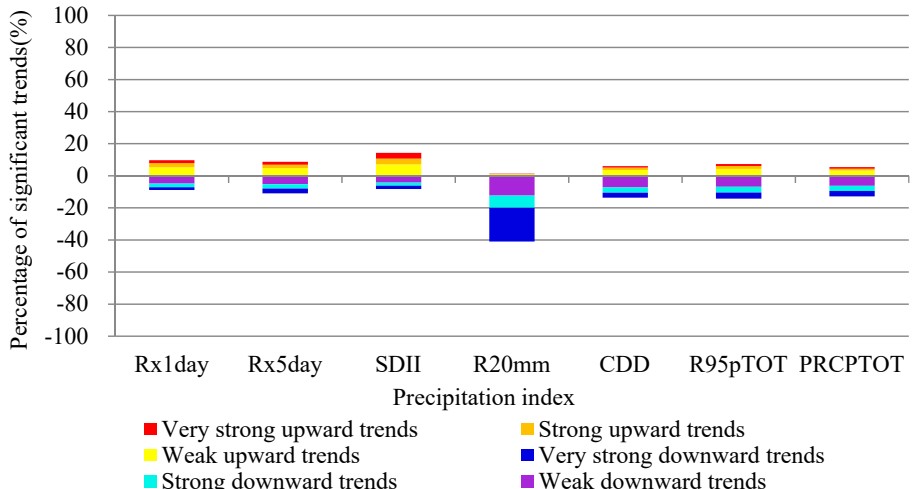

**Figure 3.** Percentages of trend (PCT) with different significant levels for each extreme precipitation index in the study area during the growth period of October 1961 to May 2016.

The percentage of downward trends is more than upward trends in all extreme precipitation indices except Rx1day and SDII (Figure 3). The percentage of downward trends is observed almost twice as frequently as that of upward trends in CDD, R95pTOT, and PRCPTOT indices, with approximately 14%, 14%, and 13%. The highest percentage of downward trends is in R20mm, which is 41%.

For all the extreme indices, the statistically significant trends (weak, strong, and very strong trends) are much more frequent than insignificant trends for four extreme temperature indices (FD0, TN10p, TN90p, and TX90p), and the percentage of insignificant trends is greater than 52%. While the insignificant trends in the other four temperature indices (GSL, TXx, TNn, TX10p) and all extreme precipitation indices dominate that period in the region.

### 3.1.2. Spatial Distributions of Trend Strengths

The spatial distribution of the trend strengths in extreme temperature indices during the growth period of October 1961 to May 2016 are shown in Figure 4. For each index, the M–K trend test was carried out for every 30-growth-period at each meteorological station. The type of trend strength which takes the majority part among the results of all 30-growth-period was treated as the value of a certain station and shown in Figure 4 (the same method was used for the drawing of Figure 5).

For the cold extreme indices FD0 and TN10p, most sites in the study area show a very significant downward trend. And the very significant downward trend in TX10p is mainly distributed in the northeastern part of the study area. In contrast to the above three extreme cold indicators, TNn has a small number of sites showing an upward trend and the distribution of sites is more dispersed. In the hot extreme index, the site with a significant upward trend in TN90p is mainly distributed in the northern winter wheat planting area and the northeast of the southern winter wheat planting area. The very significant upward trends recorded in TX90p mainly appear near the dividing line and coastal areas. For the GSL, the stations in the northern winter wheat planting areas and the north of the southern winter wheat planting areas show an upward trend.

The spatial distribution of the trend strengths in extreme precipitation indices during the growth period of October 1961 to May 2016 are shown in Figure 5. Only some of the sites in R20mm show a significant downward trend, mainly in the northern winter wheat growing area and the northwest of the southern winter wheat planting areas. The overwhelming majority of stations in other extreme precipitation indices show insignificant trends.

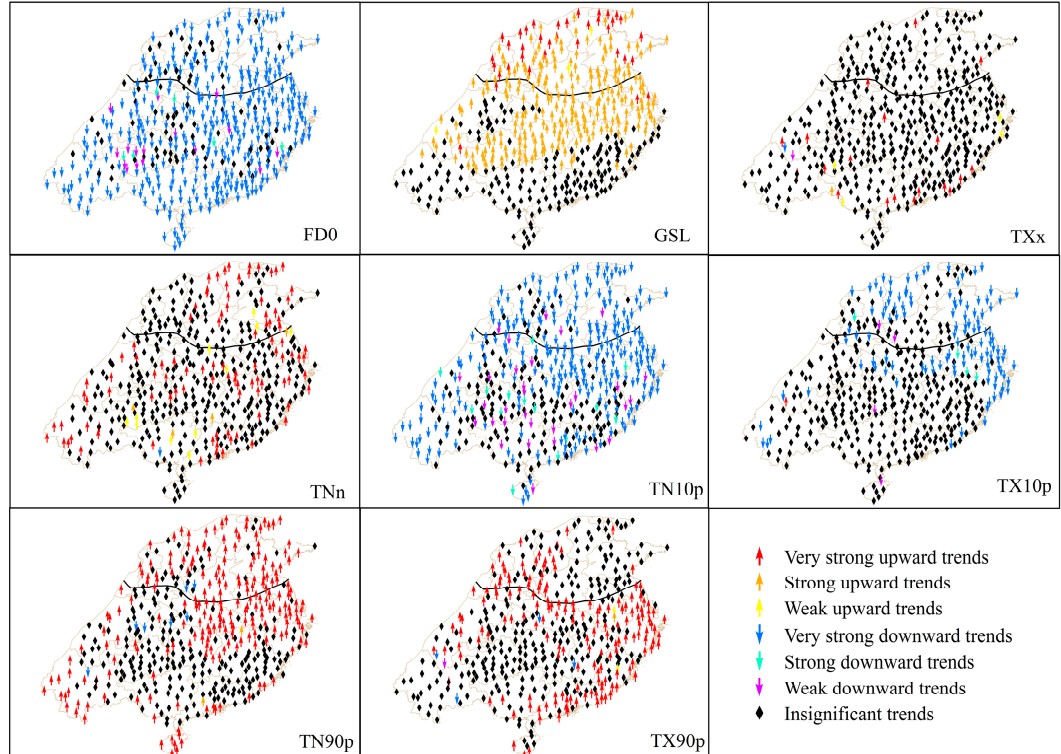

**Figure 4.** Spatial distributions of the trend strengths for the extreme temperature indices in the study area during the growth period of October 1961 to May 2016.

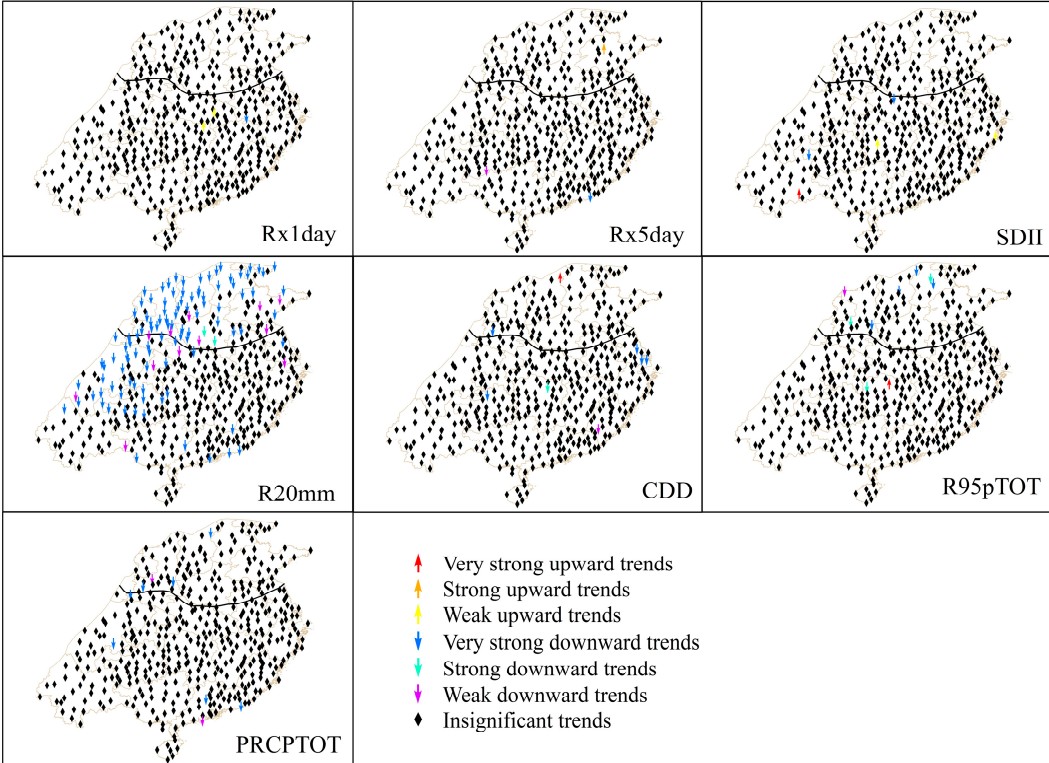

**Figure 5.** Spatial distributions of the trend strengths for the extreme precipitation indices in the study area during the growth period of October 1961 to May 2016.

### 3.2. Stability of Trends for the Extreme Indices

In this section, the stability (calculated by Equation (2)) of the significant trends (including very strong and strong) in extreme precipitation and temperature is calculated. Figure 6 and Figure 8 depict the spatial distributions of the stability respectively for the extreme temperature and precipitation indices during the growth period of October 1961 to May 2016. Figure 7 and Figure 9 show the percentages of the meteorological stations with the stability of the trends (upward or downward trends) in extreme temperature and precipitation indices.

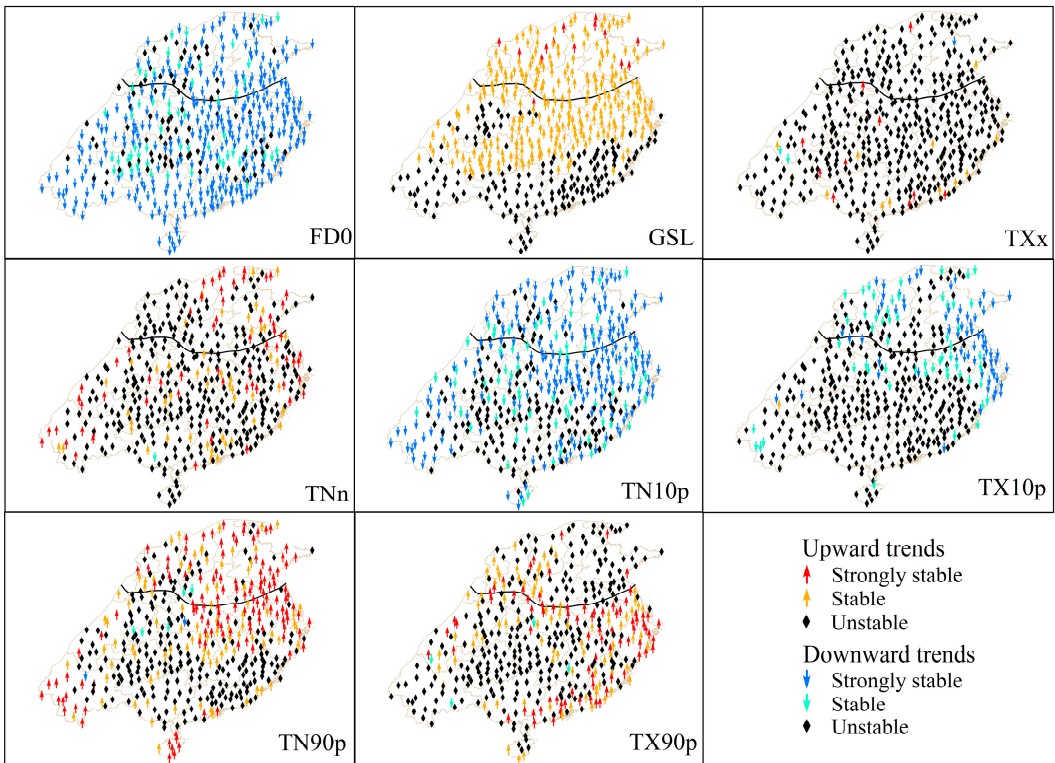

**Figure 6.** Stabilities of the statistically significant trends at each meteorological station for extreme temperature indices during the growth period of October 1961 to May 2016.

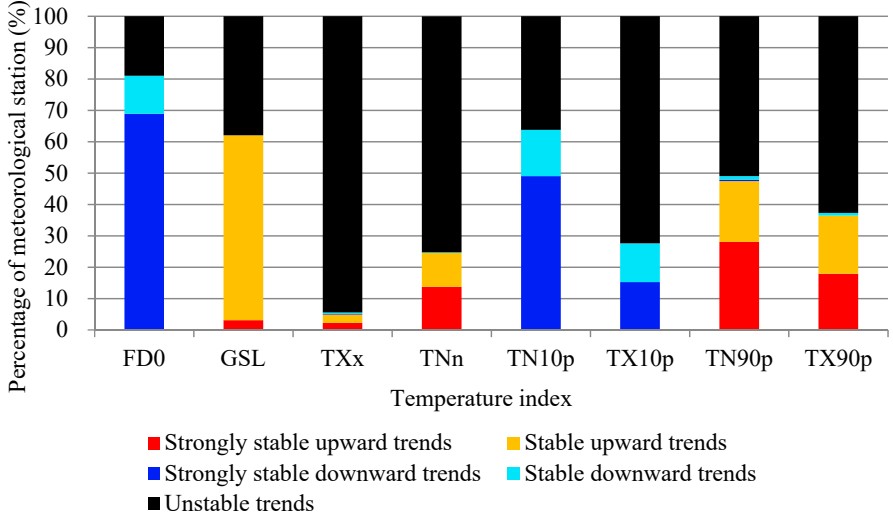

**Figure 7.** Percentage of stations with stable trends in extreme temperature indices during the growth period of October 1961 to May 2016.

### 3.2.1. Stability of Trends in Extreme Temperature

From Figures 6 and 7, the cold extreme indices FD0 and TN10p showed a stable downward trend in most of the sites in the study area, with approximately 81% in FD0 (69% of very strongly stable downward trends) and 64% in TN10p (49% of very strongly stable downward trends). Only 27% of stations in TX10p show a stable decreasing trend (strongly stable decreasing trends and stable decreasing trends), and the station is mainly distributed in the northern winter wheat planting areas and the northeast of the southern winter wheat planting areas. While in the cold extreme index TNn, the stable upward trends are found in 25% of stations in the study area, and the site distribution is relatively scattered. Hot extreme indices (TXx, TN90p, and TX90p) and GSL indicate general stable increasing trends in the period, with approximately 5% in TXx, 47% in TN90p, 36% in TX90p, and 62% in GSL. For the site distribution of the stable upward trend, the sites in TN90p are mainly distributed in the northern winter wheat planting area and the eastern part of the research area, the stations in TX90p are mainly located in the dividing line and coastal areas, the sites in GSL are located in the northern winter wheat planting area and the north of the southern winter wheat planting area.

From the above analysis, it is known that the spatial distribution characteristics of stability of trends in extreme temperature index are similar to those of the trend strengths. Most of the sites also have a stable upward (downward) trend when they have a significant upward (downward) trend.

### 3.2.2. Stability of Trends in Extreme Precipitation

As shown in Figures 8 and 9, the unstable trend in Rx1day, Rx5day, SDII, CDD, R95pTOT, and PRCPTOT dominate during the growth period of October 1961 to May 2016 in the study area, with more than 98% of stations in the six extreme precipitation indices. Only 24% of the sites in R20mm show a stable downward trend, mainly in the northern winter wheat growing area and the northwest of the southern winter wheat planting areas. From the above analysis, it is known that the spatial distribution characteristics of stability of trends in extreme precipitation index are similar to those of the trend strengths. The trend of extreme precipitation index in most sites in the study area is insignificant and unstable.

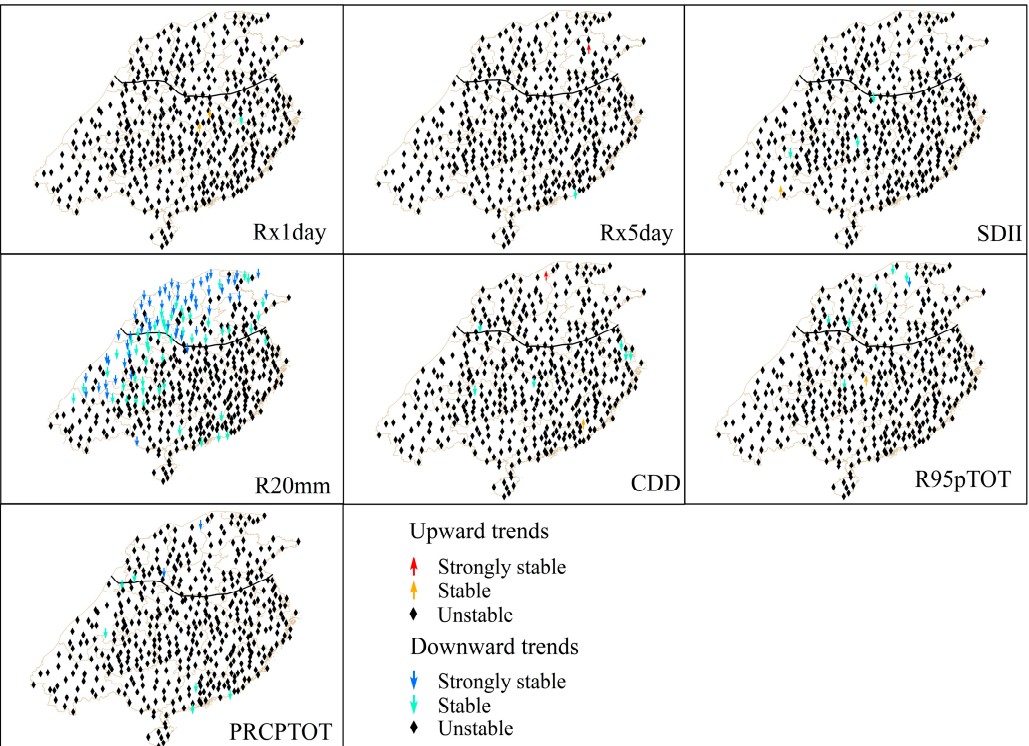

**Figure 8.** Stabilities of the statistically significant trends at each meteorological station for extreme precipitation indices during the growth period of October 1961 to May 2016.

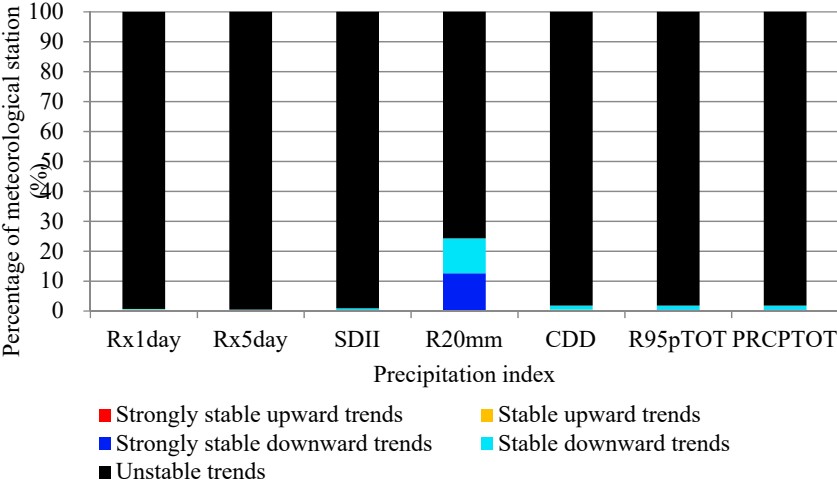

**Figure 9.** Percentage of stations with stable trends in extreme precipitation indices during the growth period of October 1961 to May 2016.

### 3.3. Trend Magnitude

Sections 3.1 and 3.2 describe the trend strengths and stability of the trends in extreme climate indices. In this section, the Sen's slope estimator method is used to analyze the spatial variability of the trend magnitudes for extreme temperature and precipitation indices. For one extreme index of a site, the following three steps were used to calculate the trend magnitude: ① Equations (3), (4) and (5) were used to calculate the Sen's slope for all the 30-growth-period moving periods during the growth period of October 1961 to May 2016 [44]. ② Then the mean value of all Sen's slope values (obtained in ①) was calculated. ③The average trend magnitude of the 30-year scale of TXx and TNn is equal to 30 times the value (obtained in ②), and the unit of the trend magnitude of these two indices is °C/30 years. The other indices trend magnitude is expressed as percentages of average values of the indices from the period October 1971 to May 2001 [6,28,44].

#### 3.3.1. Trend Magnitude in Extreme Temperature

Figures 10 and 11 show the spatial variability of the trend magnitude of the extreme temperature index. The main impression is that significant warming trends are common in most parts of the region.

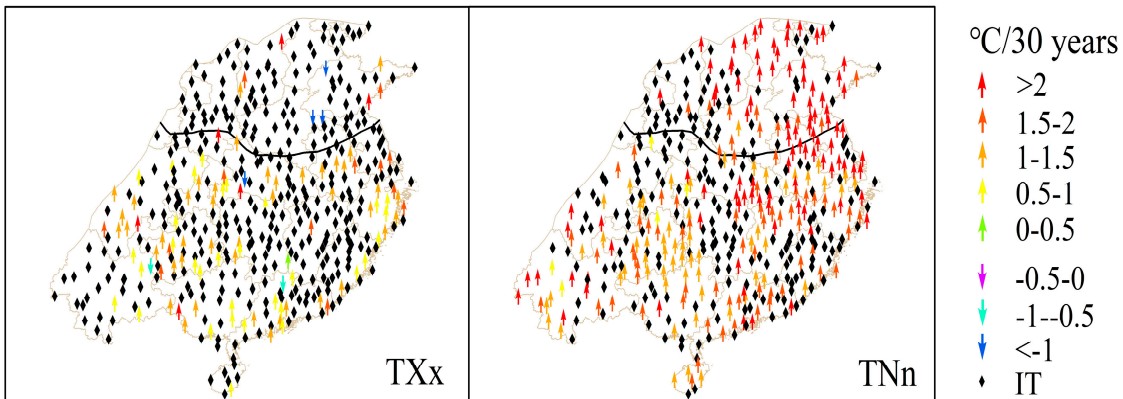

**Figure 10.** Spatial distributions of the trend magnitudes for TXx and TNn at a 30-year scale during the growth period of October 1961 to May 2016. IT represents insignificant trends, and the trend magnitude fails to pass the significance test of $p = 0.05$. The colored arrows indicate that the trend magnitude has passed the significance test of $p = 0.05$.

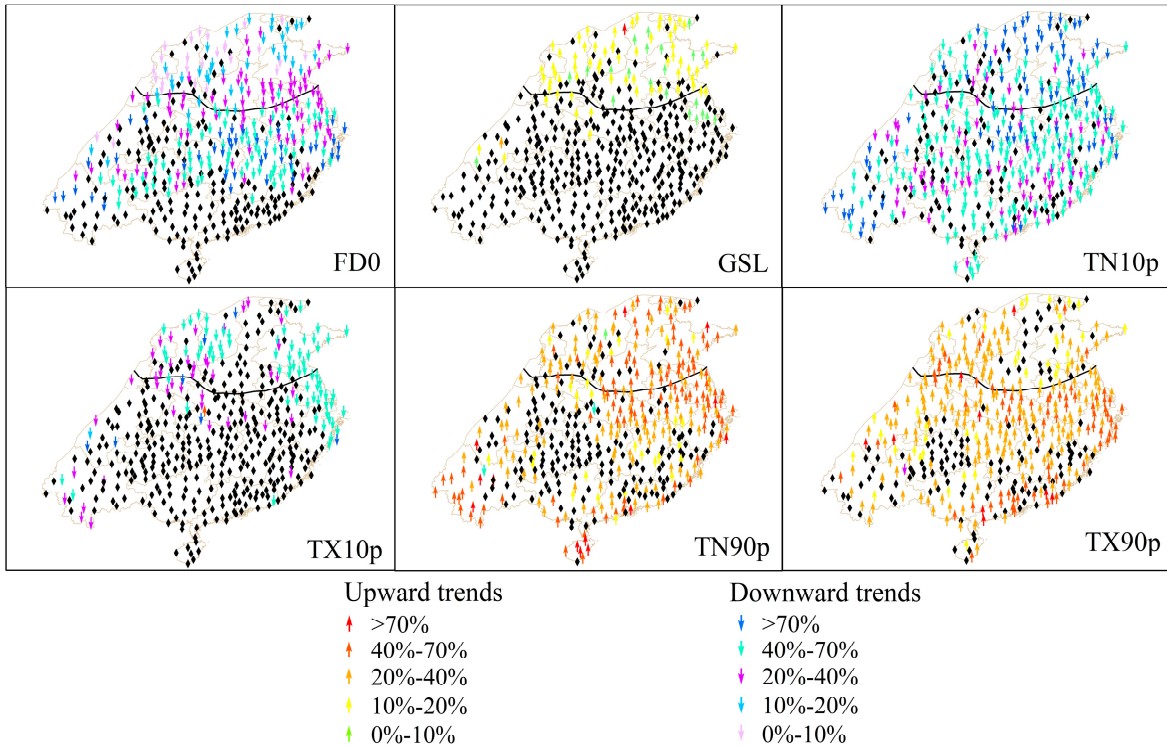

**Figure 11.** Spatial distributions of the trend magnitudes for the extreme temperature indices at a 30-year scale during the growth period of October 1961 to May 2016. The black rhombus represents insignificant trends, and the trend magnitude fails to pass the significance test of $p$ = 0.05. The colored arrows indicate that the trend magnitude has passed the significance test of $p$ = 0.05.

In Figure 10, 21% of the sites in TXx show upward trends, and 14% of stations with the upward trend magnitudes exceed 1 °C/30 years, which are mainly found in the southern winter wheat planting areas. For TNn, it can be clearly seen from the figure that the level of temperature rise is higher than that of TXx, where the temperature of 49% of the stations rises by 1 °C/30 years. Furthermore, we found that the largest warming trends (≥2 °C/30 years) predominantly occur in northern winter wheat planting area and the northeast of the southern winter wheat planting area, with 21% of sites.

For the cold extreme indices FD0 and TN10p in Figure 11, it can be clearly seen that most of the stations (more than 60% of the stations) show a downward trend. Trend magnitudes exceeding 20% are found in 45% and 77% of stations, respectively. In addition, for FD0, magnitudes exceeding 70% and between 40–70% are shown in 9% and 18% of stations, and the stations are mainly distributed in the southern winter wheat growing areas. Trend magnitudes exceeding 70% and between 40–70% in TN10p are found with 18% and 43% of sites, and the site is evenly distributed in the study area. For TX10p, the trend magnitude of 73% stations fails to pass the significance test ($p$ = 0.05), but magnitude exceeding 20% is shown in 26% of stations which are mainly distributed in the northern winter wheat growing areas and the northeast part of the southern winter wheat growing area. For hot extreme indices TN90p and TX90p, respectively 60% and 71% of the sites show an increasing trend. Trend magnitudes between 20–40% are found in 26% and 44%, respectively, and magnitudes of 40%–70% are found in respectively 23% and 16% of stations. Only 13 and 9 stations show the largest warming trends (≥70%) respectively in TN90p and TX90p in the study area. Sites showing an upward trend are evenly distributed in the study area. And the sites whose trend magnitude did not pass the significance test ($p$ = 0.05) are mainly distributed in the southern winter wheat planting areas. For GSL, the trend magnitude of 81% sites shows insignificant trends, and the stations are mainly located in the southern winter wheat growing area. Only 19% of the stations show an increasing trend with the trend magnitude of 0–20%.

### 3.3.2. Trend Magnitude in Extreme Precipitation

Figure 12 shows the spatial distributions of extreme precipitation trend magnitudes. Among the extreme precipitation indices, the trend magnitude of most sites (more than 82% of the stations) does not pass the significance test ($p = 0.05$). Only 4%, 5%, 16%, and 6% of sites in Rx1day, Rx5day, SDII and R95pTOT show upward trends, respectively. Moreover, the stations are mainly distributed in the southern winter wheat planting area. However, 7% of the sites in PRCPTOT show a downward trend. The trend of R20mm and CDD is not obvious. In general, the trend magnitude of extreme precipitation indices in the study area is not significant.

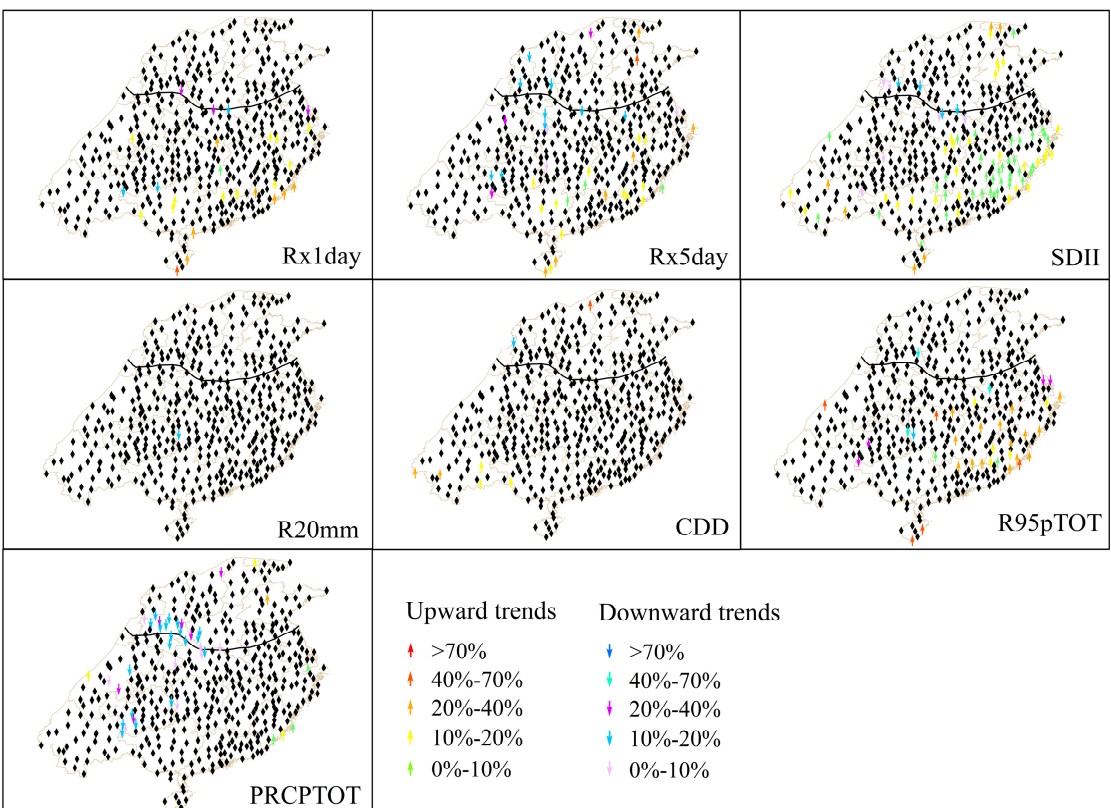

**Figure 12.** Spatial distributions of the trend magnitudes for the extreme precipitation indices at a 30-year scale during the growth period of October 1961 to May 2016. The black rhombus represents insignificant trends, and the trend magnitude fails to pass the significance test of $p = 0.05$. The colored arrows indicate that the trend magnitude has passed the significance test of $p = 0.05$.

### 3.4. Average Index Time Series

In the previous analysis, the trend strengths, stability, and trend magnitude in each site were analyzed. In this section, areal values of the extreme temperature and precipitation indices over the whole study area were calculated by averaging the respective values of all the meteorological stations. We divided the study area into two regions, 'a' and 'b', where 'a' represents the northern winter wheat planting area and 'b' represents the southern winter wheat planting area. Figures 13 and 14 show the arithmetic average of the index for all sites in the regions and show the trend strengths and trend stability test results for the arithmetic average of extreme temperature and precipitation indices for the winter wheat growth periods of October 1961 to May 2016.

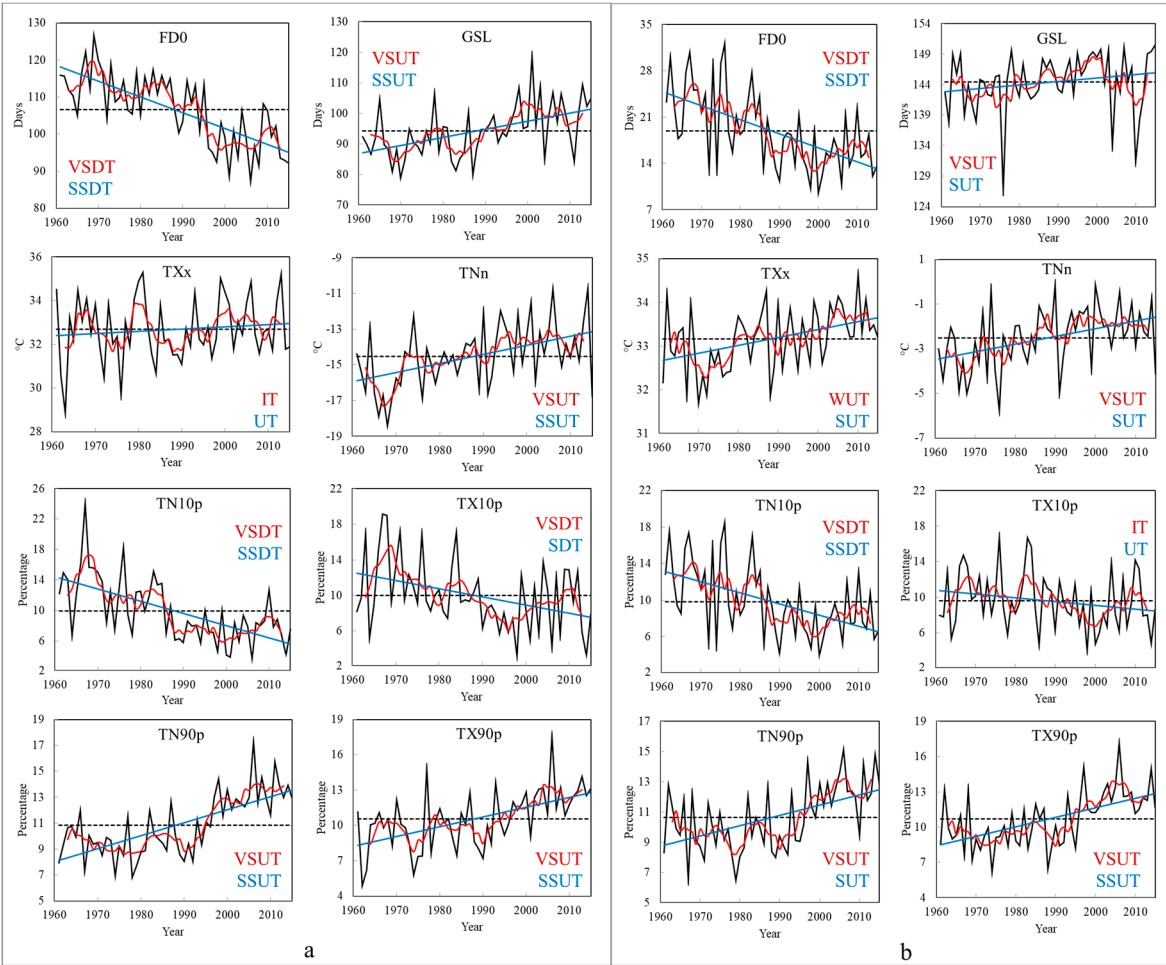

**Figure 13.** Annual variations of the spatially averaged extreme temperature indices and the trend strengths and trend stability test results during the growth period of October 1961 to May 2016. The black curves show the arithmetic average values of the total meteorological stations. The red curves give the five-year moving average. The black horizontal dotted lines represent the average level, and the blue lines show the trends. The northern China winter wheat region is denoted by '**a**'. The southern China winter wheat region is denoted by '**b**'. The trend strength is expressed in red letters. The stability of the trend is expressed in blue letters. VSDT represents very strong downward trends. VSUT represents very strong upward trends. WUT represents weak upward trends. IT represents insignificant trends. SSDT represents strongly stable downward trends. SSUT represents strongly stable upward trends. UT represents unstable trends. SDT represents stable downward trends. SUT represents stable upward trends.

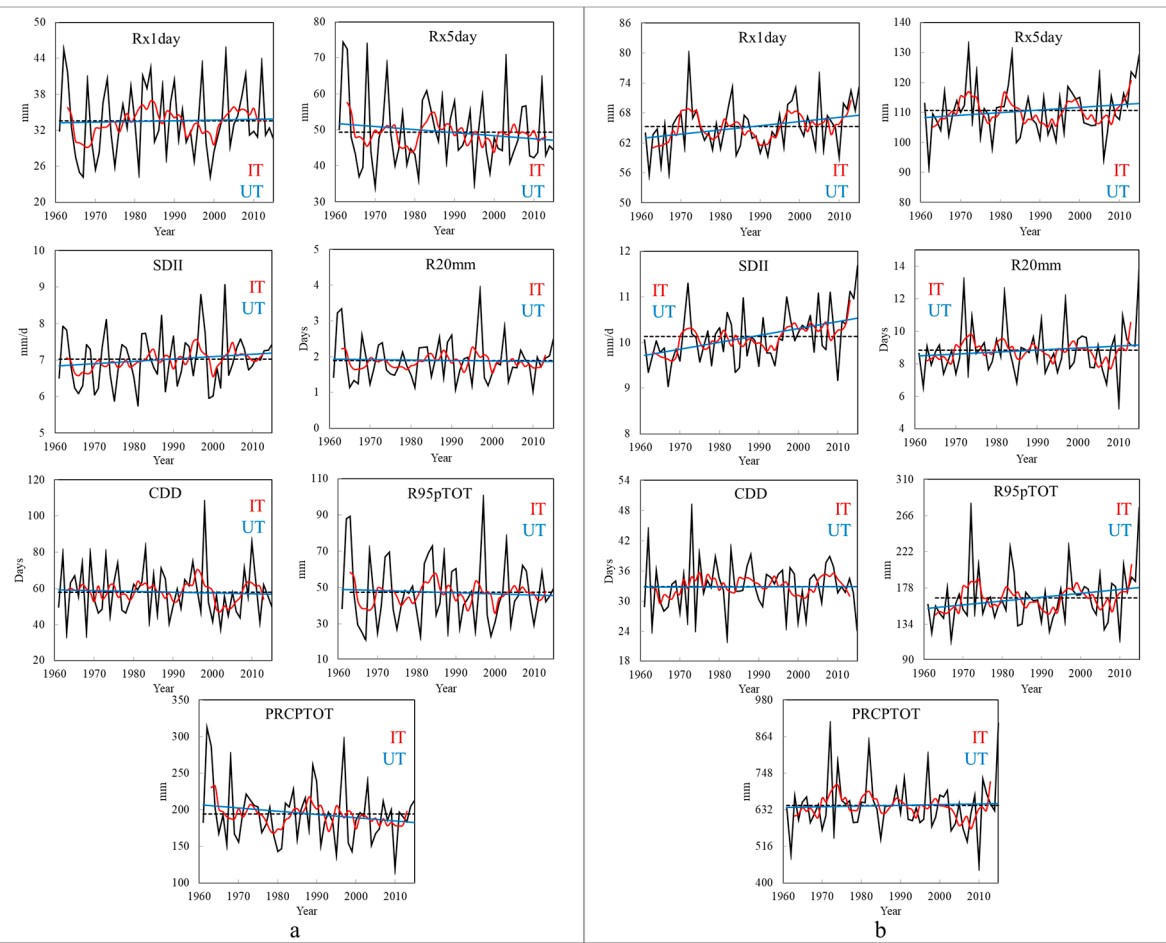

**Figure 14.** Annual variations of the spatially averaged extreme precipitation indices and the trend strengths and trend stability test results during the growth period of October 1961 to May 2016. The black curves show the arithmetic average values of the total meteorological stations. The red curves give the five-year moving average. The black horizontal dotted lines represent the average level, and the blue lines show the trends. The northern China winter wheat region is denoted by (**a**). The southern China winter wheat region is denoted by (**b**). The trend strength is expressed in red letters. The stability of the trend is expressed in blue letters. IT represents insignificant trends. UT represents unstable trends.

### 3.4.1. Extreme Temperature Indices

As can be seen from Figure 13, the cold extreme indices FD0 and TN10p in the north and south winter wheat planting areas show a very strong and strongly stable downward trend. The very strong and stable downward trend is found in TX10 in the northern winter wheat planting area, while an insignificant trend is found in the southern area. TNn, in contrast to the above cold extreme indices, shows a very significant upward trend in the two winter wheat planting areas, and a very stable downward trend in the northern winter wheat planting area. In the hot extreme index, all the indices (TXx, TN90p and TX90p) of the two winter wheat planting areas show an upward trend, and the trends in TN90p and TX90p are very significant, especially for TX90p which has significant and strongly stable trends. GSL shows the same trend strengths (very strong upward trend) in both regions, and trend stability in the northern winter wheat planting area is even better. The changing trend of all these extreme temperature indices indicates that the climate warming process has occurred, and extreme temperature events tend to be more frequent during the growth period of the winter wheat in the study area.

3.4.2. Extreme Precipitation Indices

Figure 14 reveals that only the indices of Rx1day and SDII are on increasing trends in both two winter wheat planting areas (denoted by the blue lines in Figure 14), and the changing trend of all extreme precipitation indices is insignificant and unstable. In the northern winter wheat planting area, all of the indices (except Rx1day and SDII) show a slight decline trend, in which the trend of Rx5day and PRCPTOT decreases obviously relative to other indices. However, in the southern winter wheat planting area, the upward trend is found in all of the indices (except Rx1day and SDII), where the increasing trend of Rx5day and R95pTOT is more obvious. All these results indicate that although there is an inter-annual variation of rainfall in the growth period of winter wheat for the 56 growth periods (from October 1961 to May 2016), the frequency of extreme precipitation events is not increased significantly.

## 4. Discussion

The spatial and temporal distribution of the trend strengths, trend stability and trend magnitudes are revealed through the previous analysis.

The analysis results of extreme temperature indices are: (1) the results of cold extreme indices (FD0, TNn, TN10p, and TX10p) show that the frost days (when daily minimum temperature is <0 °C) is decreasing, the minimum of daily minimum temperature is increasing, the number of the cool nights and the cool days is decreasing. (2) The hot extreme indices (TXx, TN90p, and TX90p) analysis shows that the maximum of daily maximum temperature is increasing, the number of warm nights and warm days is increasing. (3) And the results of GSL analysis show that the growing season length (when the daily average temperature is >5 °C) is increasing during the growth period of the winter wheat in the study area. These results show that the study area is undergoing a process of warming during the growth period of the winter wheat, which will shorten the growth period of winter wheat and lead to a decline in the yield [50,51]. In this paper, the study area involves three major economic circles in China (showing in Figure 1). From Tables 2 and 3, for the extreme cold index FD0, more than 90% of the sites in the three major economic circles show a very strong and strongly stable downward trend, while only about 60% of the sites in the other regions (the other regions refer to the rest of the study area except the three major economic circles) show the trend. Additionally, the same trend can be found in TN10p, with approximately 80% of the stations in the three economic circles and about 40% of the stations in other regions. For the hot extreme index TN90p, the very strong and strongly stable upward trends are tested in about 50% of stations in the three major economic circles and 23% of stations in the other regions. Similarly, for TX90p, the percentage of sites showing a very strong and strongly stable upward trend in the three major economic circles is larger than that in the other regions. From Figures 4, 6, 10 and 11, it can also be seen that the warming trend of the sites in the three major economic circles is more obvious and stable than that in the other regions. This phenomenon may be related to the urban heat island (UHI) effect formed by the large-scale urban agglomerations in the three major economic circles, and IPCC [52] pointed out that the UHI effects were real but local. The second reason is that the change of atmospheric circulation affected heat transportation in the area [29,53]. In winter, the southwesterly wind has become strengthened and weakened the Siberian northwest monsoon, resulting in the reduction of extreme cold events. The third reason may be an increase in greenhouse gas emissions caused by coal-fired heating in the north of the Huaihe River in China.

The analysis of the extreme precipitation indices shows that the trend of most indices (Rx1day, Rx5day, SDII, CDD, R95pTOT, and PRCPTOT) is not obvious. Only 24% of the sites in R20mm show a strong and stable downward trend, mainly in the northern winter wheat growing area and the northwest of the southern winter wheat planting areas. The winter wheat growth period is from October to May. Therefore, the precipitation data analyzed in this paper do not contain the data of the flood season (usually from June to September), while most events and amounts of the precipitation occur in flood season with heavy rainfall intensity and long rainfall duration [14,54]. Moreover, a previous study shows that the increasing trends of the precipitation

and three wet indices (R95pTOT, CWD and CDD) mainly occurred in summer (June to August) and winter (December to February), and the decreasing trends occurred in spring (March to May) and autumn (September to November) [55,56]. Contrary to the trend of the precipitation and three wet indices, the dry index CDD shows a downward trend in summer and winter, and an upward trend in spring and autumn [55]. The winter wheat growth period includes winter, autumn and spring, which may lead to insignificant changing trends in the extreme precipitation indices (except R20mm) during the study period. The changing trend of R20mm in the study area may be related to the reduction of extreme precipitation events in the central-north China (involving northern winter wheat growing areas) during both spring and autumn and the reduction of extreme precipitation events in northwest China (involving the northwest of the southern winter wheat planting areas) during spring, winter, and autumn [57]. According to the meteorological factors, the precipitation in the study area is mainly influenced by the East Asian monsoon, the El Nino-Southern Oscillation, the change of sea surface temperature in the Pacific Ocean and the Pacific Decadal Oscillation [14,43,58]. It is also possible that the aerosol changes caused by human activities which potentially impact the process of vapor transition and vapor condensation [14,43,58], and ultimately affect the formation of precipitation. The combined effect of all these factors led to an insignificant change in extreme precipitation indices in the study area.

**Table 2.** Percentage of stations with trends strengths in extreme temperature indices (%).

| Trend Strengths | FD0 | GSL | TXx | TNn | TN10p | TX10p | TN90p | TX90p |
|---|---|---|---|---|---|---|---|---|
| Very Strong Upward Trends | 0(0) [1] | 13(7) | 2(5) | 35(18) | 0(0) | 0(0) | 68(41) | 48(32) |
| Strong Upward Trends | 0(0) | 0(0) | 0(0) | 1(1) | 0(0) | 0(0) | 0(1) | 1(0) |
| Weak Upward Trends | 0(0) | 1(1) | 2(1) | 1(3) | 0(0) | 0(0) | 0(0) | 1(0) |
| Very Strong Downward Trends | 95(73) | 20(42) | 0(0) | 0(0) | 83(49) | 52(19) | 0(2) | 0(1) |
| Strong Downward Trends | 0(2) | 0(0) | 0(0) | 0(0) | 2(3) | 3(0) | 0(0) | 0(0) |
| Weak Downward Trends | 0(4) | 0(1) | 0(0) | 0(0) | 4(8) | 0(1) | 0(0) | 0(0) |
| Insignificant Trends | 5(21) | 66(50) | 96(94) | 62(78) | 11(40) | 45(79) | 32(56) | 49(66) |
| Total | 100(100) | 100(100) | 100(100) | 100(100) | 100(100) | 100(100) | 100(100) | 100(100) |

[1] The data outside the brackets represent the percentage of meteorological stations with different trends strengths in the three major economic circles (93 meteorological stations in the three major economic circles as shown in Figure 1). The data in brackets are the percentage of meteorological stations with different trends strengths in the other regions (360 meteorological stations in the other regions as shown in Figure 1).

**Table 3.** Percentage of stations with stable trends in extreme temperature indices (%).

| Stability of Trends | FD0 | GSL | TXx | TNn | TN10p | TX10p | TN90p | TX90p |
|---|---|---|---|---|---|---|---|---|
| Strongly Stable Upward Trends | 0(0) [1] | 4(3) | 2(2) | 27(10) | 0(0) | 0(0) | 48(23) | 35(13) |
| Stable Upward Trends | 0(0) | 10(5) | 1(3) | 11(11) | 0(0) | 0(0) | 19(19) | 15(19) |
| Strongly Stable Downward Trends | 91(63) | 16(39) | 1(0) | 0(0) | 78(41) | 35(10) | 0(1) | 0(0) |
| Stable Downward Trends | 3(14) | 4(4) | 0(1) | 0(0) | 9(16) | 19(11) | 0(1) | 0(1) |
| Unstable Trends | 5(23) | 66(50) | 96(94) | 62(79) | 13(42) | 45(79) | 32(56) | 49(66) |
| Total | 100(100) | 100(100) | 100(100) | 100(100) | 100(100) | 100(100) | 100(100) | 100(100) |

[1] The data outside the brackets represent the percentage of meteorological stations with different stable trends in the three major economic circles (93 meteorological stations in the three major economic circles as shown in Figure 1). The data in brackets are the percentage of meteorological stations with different stable trends in the other regions (360 meteorological stations in the other regions as shown in Figure 1).

## 5. Conclusions

In this paper, the following conclusions can be obtained through a detailed analysis of 15 extreme climatic indices of 56 winter wheat growing periods in winter wheat growing area of China:

(1) According to the above analysis of the extreme temperature index, from the perspective of time, the winter wheat growth period is undergoing a warming process with a decrease in extreme cold events and an increase in extreme hot events. The downward trends of the cold extreme indices FD0 and TN10p are the most obvious, with 81% and 64% of the stations showing a stable downward trend, respectively, and the sites with the trend magnitude exceeding 20% are accounted for 59% and 93%, respectively. Among the hot extreme indices, TN90p and TX90p show the most obvious upward trend,

with 47% and 36% of the sites showing an upward trend, respectively, and 58% and 72% of the sites with the trend magnitude exceeding 20%, respectively. From the perspective of space, the warming trends in the northern winter wheat planting area and north of the southern winter wheat planting area are more significant, especially in the three major economic circles. The extreme temperature index can well reflect the temperature change of the growth period of winter wheat in 56 years.

(2) The change trends of most extreme precipitation indices are insignificant and unstable in the winter wheat growth period in the study area. There are 24% of the sites in R20mm that show a strong and stable downward trend, mainly in the northern winter wheat growing area and the northwest of the southern winter wheat planting areas. Moreover, because the events and amounts of the precipitation in China mostly occur in flood season with heavy rainfall intensity and long rainfall duration, extreme precipitation indices are not suitable for evaluating climate change during the growth period of winter wheat in this region. Further studies can analyze the influence of climate change on the effective precipitation in the different growth stages of winter wheat.

**Author Contributions:** All the authors have contributed to the conception and development of this manuscript. H.N. carried out the analysis and wrote the paper. T.Q. reviewed and edited the manuscript. H.Y. conceived and designed the framework. J.C. and S.H. embellished the English expression of the thesis. Z.L. and Z.S. provided assistance in calculations and figure productions.

**Funding:** This work was supported by the National Key Research and Development Project (Grant No. 2016YFA0601503); the National Science Fund for Distinguished Young Scholars (Grant No. 51725905); the National Key Research and Development Project (Grant No. 2017YFA0605004); and the Representative Achievements and Cultivation Project of State Key Laboratory of Simulation and Regulation of Water Cycle in River Basin (Grant No. 2016CG02).

**Acknowledgments:** The basic meteorological data was provided by the National Meteorological Information Center of China Meteorological Administration.

**Conflicts of Interest:** The authors declare no conflict of interest.

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
