# Peer review of "Trend Analysis of Temperature and Precipitation Extremes during Winter Wheat Growth Period in the Major Winter Wheat Planting Area of China"

_atmosphere, doi:10.3390/atmos10050240_

Round 1

Reviewer 1 Report

This paper researches on the Trend of temperature and precipitation extremes during winter wheat growth period in the major winter wheat planting area of China, a total of 56 growth periods during 1961/10-2016/5 are taken into consideration, the spatial and temporal distribution of the trend strengths, trend stability and trend magnitudes of the study area are discussed based on the 8 temperature indices and 7 precipitation indices from 453 meteorological stations. The results further validate the effect of climate change on the wheat planting, which can help to formulate a better agronomic management practice of winter wheat. Aside from some methodology concerns, I have no major problems with the study content or approach. My recommendation is Minor Revision.

1. The author have described the data quality and homogeneity method in detail in Section 2.3, how many outlying data, unreasonable data are identified and excluded? More details about data, including total amount of data, percentage of valid and invalid data, valid and invalid meteorological stations, are suggested to be added.

2. Table 1 Lists the extreme indices of temperature and precipitation. The study period is from 1961 to 2016, but why the TN10p, TX10p, TN90p, TX90p is defined from 1971/10 to 2001/05. The following discussion is based on 1961-2016 or 1971-2001? The author should make it clear.

3. There are some qualitative descriptions such as very strong upward trends, strong downward trends, stable upward trends, and etc. how to define the degree of upward or downward trends, is there a quantitative definition? Please verify.

4. Considering that there are several climate regions in china, their climate change might have different trend and characteristics in different regions, why not discuss and compare the temperature and precipitation extreme in China's three major economic circles, the Yangtze River Delta, the Pearl River Delta, and the Beijing-Tianjin-Hebei area) in details.

5. As the author stated, the urban heat island (UHI) effect may be one of the reasons for the temperature rise, can the author choose several typical urban and suburb meteorological stations, and compare their different trend of temperature rise?

6. The winter precipitation seldom has heavy rainfall intensity and long rainfall duration, which may lead to insignificant change trends in precipitation during the study period, these trends are based on the conventional indices, could the author adopt more suitable indices for winter precipitation analysis?

7. There are some interesting findings, such as 24% of the sites in R20mm show stable downward trend, mainly in the northern winter wheat growing area and the northwest of the southern winter wheat planting areas, should be added to the Section 4, and the possible reason should be discussed.

8. There should be more quantitative summary in Discussion and Conclusions.

Author Response

Reviewer's comments:

1. The author have described the data quality and homogeneity method in detail in Section 2.3, how many outlying data, unreasonable data are identified and excluded? More details about data, including total amount of data, percentage of valid and invalid data, valid and invalid meteorological stations, are suggested to be added

Reply or modification:

I made some changes and highlighted them in section 2.2: "Due to the unavoidable occurrence of unreasonable data and missing values in the data, it is necessary to carry out quality control and homogeneity test on the data [26]."

I made some changes and highlighted them in section 2.3: ''Through data quality checks, we identified less than 0.2% observations per station as outliers, and these values were set to the multi-year average of the same day. On average, we corrected approximately 0.8%, 1.2% and 2.3% of observations per station for maximum temperature, minimum temperatures and precipitation, respectively. Through the homogeneity check of the data, 121 change points were detected among 487 stations in the monthly data series. All of the change points occurred between 1978 and 1980. And all the daily data in the change point are set missing. The stations with more than 5% data missing are excluded from analysis [26,33]. After data quality and homogeneity checking, 34 stations with missing data more than 5% are excluded and 453 out of 487 stations are thus included in this study.''

Reviewer's comments:

2. Table 1 Lists the extreme indices of temperature and precipitation. The study period is from 1961 to 2016, but why the TN10p, TX10p, TN90p, TX90p is defined from 1971/10 to 2001/05. The following discussion is based on 1961-2016 or 1971-2001? The author should make it clear.

Reply or modification:

Reply

This paper chooses the growth period of 1971/10-2001/05 as the base period, which is explained in the last sentence of the first paragraph of section 2.4. Now I add a note about the base period after "Table 1".

Here I will introduce the calculation process of these indices (TN10p, TX10p, TN90p, TX90p). Taking the calculation of TN10p as an example. Firstly, the daily minimum temperature (TN) of the base period (1971/10-2001/05) is arranged from small to large, and the TN of the 10th percentile of the base period is obtained. Then, the number of days in which the TN is less than 10th percentile in a certain growth period (e.g., 1961/10-1962/05) is counted. Finally, the number of days (TN<10th percentile of the base period) is divided by the total number of days in the growth period (243 days from 1961/10 to 1962/05) to obtain TN10p. The TN10p of all 55 growth periods (1961/10-2016/05) is obtained by the same calculation for other growth periods.

Modification:

I made some changes and highlighted them in section 2.4: ''The period of 1971/10-2001/05 is the base period.''

Reviewer's comments:

3. There are some qualitative descriptions such as very strong upward trends, strong downward trends, stable upward trends, and etc. how to define the degree of upward or downward trends, is there a quantitative definition? Please verify.

Reply or modification:

Reply:

In section 2.5.1 of this paper, the trend strength of extreme indices is defined as follows (p is the tested significance level): Very strong trends: p≤0.05. Strong trends: 0.05<p≤0.1. Weak trends: 0.1<p≤0.2. Insignificant trends: p>0.2. The normally distributed statistics with the significance levels of 0.05, 0.1, and 0.2 are 1.96, 1.65, and 1.28, respectively. When the statistic obtained by M-K test is greater than or equal to 1.96, it shows a very strong upward trend. When the statistic obtained by M-K trend test is less than or equal to -1.96, it shows a very strong downward trend. When the statistic obtained by M-K trend test is greater than or equal to 1.65 and less than 1.96, it shows a strong upward trend. By analogy, we get the trend strength of extreme indices, which provide a qualitative representation of the trends of extreme climate indices. Then the ''3.3. Trend magnitude'' of the paper is a quantitative description of the trend of extreme climate indices, in which TXx and TNn are expressed in terms of the increase of degrees Celsius every 30 years, while other extreme indices are expressed in terms of the increase of percent every 30 years.

Reviewer's comments:

4. Considering that there are several climate regions in china, their climate change might have different trend and characteristics in different regions, why not discuss and compare the temperature and precipitation extreme in China's three major economic circles, the Yangtze River Delta, the Pearl River Delta, and the Beijing-Tianjin-Hebei area) in details.

5. As the author stated, the urban heat island (UHI) effect may be one of the reasons for the temperature rise, can the author choose several typical urban and suburb meteorological stations, and compare their different trend of temperature rise?

Reply or modification:

Considering the relevance of the three major economic circles and the UHI effect involved in these two opinions, we will reply to them together. Tables 4 and 5 are added in the fourth section of the paper, and the trend significance and stability of the sites in the three major economic circles (the large-scale urban agglomerations are formed in the areas) and the other regions are counted. In addition, relevant descriptions and discussions are added in the first paragraph of the fourth section, which are shown in bold. Since there are many changes, we won't show them here. Please check them in the article.

Reviewer's comments:

6. The winter precipitation seldom has heavy rainfall intensity and long rainfall duration, which may lead to insignificant change trends in precipitation during the study period, these trends are based on the conventional indices, could the author adopt more suitable indices for winter precipitation analysis?

7. There are some interesting findings, such as 24% of the sites in R20mm show stable downward trend, mainly in the northern winter wheat growing area and the northwest of the southern winter wheat planting areas, should be added to the Section 4, and the possible reason should be discussed.

Reply or modification:

Considering the relevance of the winter precipitation and the R20mm involved in these two opinions, we will reply to them together. Relevant descriptions and discussions are added in the second paragraph of the fourth section, which are shown in bold. Since there are many changes, we won't show them here. Please check them in the article.

Reviewer's comments:

8. There should be more quantitative summary in Discussion and Conclusions.

Reply or modification:

I made some changes and highlighted them in the second paragraph of section 5: ''The downward trends of the cold extreme indices FD0 and TN10p are the most obvious, with 81% and 64% of the stations showing a stable downward trend respectively, and the sites with the trend magnitude exceeding 20% are accounted for 59% and 93% respectively. Among the hot extreme indices, TN90p and TX90p are showed the most obvious upward trend, with 47% and 36% of the sites showing an upward trend respectively, and 58% and 72% of the sites with the trend magnitude exceeding 20% respectively.''

I made some changes and highlighted them in the third paragraph of section 5: ''There are 24% of the sites in R20mm show strong and stable downward trend, mainly in the northern winter wheat growing area and the northwest of the southern winter wheat planting areas.''

New references have been added in the ''References'' section

Reviewer 2 Report

Review for “Trend analysis of temperature and precipitation 2 extremes during winter wheat growth period in the 3 major winter wheat planting area of China”

This is an interesting piece of study looking at trends in extreme temperature and precipitation metrics in China from historical observations. I find the research design is good and the result clearly presented in the paper. However, there is a very serious mistake related to the significance level and definition of trend. It has to be rectified. in I have provided detailed reasons as followed detailed comments. Therefore, I suggest an accept with major review for the current version. Authors have to correct those sections before the paper is appropriate for publication in the Atmosphere.

Line 41: “China is one region which with obviously climate change in the world.” Please fix the sentence.

Figure 4: The map for GSL is showing a very interesting clear divide in northern, central and southern China, please provide more specific explanation for this pattern.

In section 3.2.1, 3.3.1, 3.3.2, the analysis includes all very strong, strong, weak, and even insignificant trend. The author has already used a very loose statistic significance level for the strong and weak trend, normally people use the significance level of 0.05 for all their tests. It makes no sense to include trends with > 0.2 significance in those two sections. Because if it is not statistically significant, we can not say there is a trend. The analysis for those two sections has to be completely redone. Please remove all insignificant trends and do the same analysis.

Author Response

Reviewer's comments:

1. Line 41: “China is one region which with obviously climate change in the world.” Please fix the sentence.

Reply or modification:

I made some changes and highlighted them in section 1: "In China, the changes in extreme climate are uneven both in space and time because of the large landmass and various climatic factors [6], and the extreme climate changes are crucial to the local population, economy and environment [7]"

Reviewer's comments:

2. In section 3.2.1, 3.3.1, 3.3.2, the analysis includes all very strong, strong, weak, and even insignificant trend. The author has already used a very loose statistic significance level for the strong and weak trend, normally people use the significance level of 0.05 for all their tests. It makes no sense to include trends with > 0.2 significance in those two sections. Because if it is not statistically significant, we can not say there is a trend. The analysis for those two sections has to be completely redone. Please remove all insignificant trends and do the same analysis.

Reply or modification:

Reply:

Globally, the frequency of extreme temperature and precipitation events is gradually increasing. In order to describe the trend of this disastrous event in detail and capture its extremely subtle change trend, in section 2.5.1 of this paper, the trend is divided into four levels with reference to 4 literatures [6,28,44,45], namely: Very strong trends: p≤0.05. Strong trends: 0.05<p≤0.1. Weak trends: 0.1<p≤0.2. Insignificant trends: p>0.2. (p is the testing significance level). I added relevant literature and instructions in section 2.5.1 of the article. In section 3.1, the percentage and spatial distribution of extreme climate indices in different strengths in the study area are described.

In general, the trend of hydrological or meteorological sequence data is usually detected on the basis of a specific time period, and the test results are strongly influenced by the selected time periods. In order to eliminate the instability of the trend test, the concept of trend stability is introduced in this paper. The stability of such trends is expressed as a percentage of the 30-growth-period during which these trends were statistically significant (p≤ 0.1). That is to say, for a station, we use M-K trend test to calculate the trend of 26 movable 30-growth-period during 1961/10-2016/05. When the number of times of very strong increasing trend (p≤ 0.05) and strong increasing trend (0.05<p≤0.1) is greater than 16 (i.e. more than 60% of 26), we can judge that the change trend of the station is very stable. This is explained in detail in section 2.5.2, and the specific results are analyzed in section 3.2.

The previous analysis is only a qualitative description of the extreme climate indices. In order to quantitatively describe the changes of extreme climate indies, the slope of 26 30-growth-period of extreme climate indices is calculated by Sen's slope estimator, and the gradient is arranged in order from small to large, from which the median value is selected as the trend magnitude. According to relevant research, the slope computed by Sen’s slope estimator is a robust estimate of the magnitude of a trend, which has been widely used in identifying the slope of the trend line in hydrological and meteorological time series [26,47-49]. Detailed description and analysis can be found in sections 2.5.3 and 3.3 of the papper.

Modification:

I made some changes and highlighted them in section 2.5.1: ''Trends were identified as statistically significant when they were significant at p ≤0.1, but trends at p ≤0.2 were also taken into consideration and designated as weak trends [44,45]. Therefore, the classification criteria of the trend of extreme climate indices can be divided by referring to relevant studies [6,28]: Very strong trends: p≤0.05. Strong trends: 0.05<p≤0.1. Weak trends: 0.1<p≤0.2. Insignificant trends: p>0.2. (p is the tested significance level)''

I made some changes and highlighted them in section 2.5.2: ''In general, the trend of hydrological or meteorological sequence data is usually detected on the basis of a specific time period, and the test results are strongly influenced by the selected time periods [6,44]. For the same elements, it is likely that different trends result from different time periods, which reflects the instability of the trend [6]. In order to eliminate the instability of the trend test, the trend of the 30-growth-period (26 movable 30-growth-period during 1961/10-2016/05) of extreme climate indices is calculated by M-K trend test. The stability of such trends is expressed as a percentage of the 30-growth-period during which these trends were statistically significant (p≤ 0.1). That is to say, we need to count the number of very strong upward (or downward, p≤ 0.05) and strong upward (or downward, 0.05<p≤0.1) trends in the 30-growth-period sequence, and then calculate the percentage of them in the total number M. The trend stability of each extreme index can be determined by the following equation [44]:

I made some changes and highlighted them in section 2.5.3: ''According to relevant research, the slope computed by Sen’s slope estimator is a robust estimate of the magnitude of a trend, which has been widely used in identifying the slope of the trend line in hydrological and meteorological time series [26,47-49].''

New references have been added in the ''References'' section

Round 2

Reviewer 2 Report

1. For Figure 10 and 11, I still believe you should only include trends that are only statistically significant (even with the very weak significance level at 0.2). In the most original Lupikasza 2009 paper, it only reports trends that passed the 0.2 significance test. Otherwise those magnitudes are meaningless.

2. Figure 13 and Table 2, I think you should remove table 2 and report the significance level and stability index in solid numbers directly on each of the subplot in Figure 13. Right now table 2 is redundant and not very effective in communicating the information. Same thing the Figure 14.

3. I still believe any trend that can not pass 0.1 significance level from MK test can not be categorized as a strict trend. Playing around with statistical significance (especially loose them) is not an very interesting and innovative scientific practice. Copycat it again and again is even worse.

You have to address 1 and 2 to make this work publishable. Thank you.

Author Response

Reviewer's comments:

1. For Figure 10 and 11, I still believe you should only include trends that are only statistically significant (even with the very weak significance level at 0.2). In the most original Lupikasza 2009 paper, it only reports trends that passed the 0.2 significance test. Otherwise those magnitudes are meaningless.

Reply or modification:

We performed a significant test on the trend magnitude (p=0.05), redrawn Figures 10, 11 and 12, and changed the description. Detailed descriptions and discussions are added in section 3.3, which are shown in bold. Since there are many changes, we won't show them here. Please check them in the article. Thank you.

Reviewer's comments:

2. Figure 13 and Table 2, I think you should remove table 2 and report the significance level and stability index in solid numbers directly on each of the subplot in Figure 13. Right now table 2 is redundant and not very effective in communicating the information. Same thing the Figure 14.

Reply or modification:

We removed Tables 2 and 3 and added the information expressed in the tables to Figures 13 and 14. Detailed descriptions and discussions are added in section 3.3, which are shown in bold. Since there are many changes, we won't show them here. Please check them in the article. Thank you.

Reviewer's comments:

3. I still believe any trend that can not pass 0.1 significance level from MK test can not be categorized as a strict trend. Playing around with statistical significance (especially loose them) is not an very interesting and innovative scientific practice. Copycat it again and again is even worse.

Reply or modification:

Here, in order to describe the trend of extreme climate in detail and capture its extremely subtle change trend, we have used the research results of the predecessors to divide the trend level. Your comments are crucial to our future work, and we must pay attention to it in future scientific research. Thank you.

Round 3

Reviewer 2 Report

The authors made significant changes to their manuscript and I agree that this version is suitable for publication at "Atmosphere"